# Human Albumin Infusion in Critically Ill and Perioperative Patients: Narrative Rapid Review of Meta-Analyses from the Last Five Years

**DOI:** 10.3390/jcm12185919

**Published:** 2023-09-12

**Authors:** Christian J. Wiedermann

**Affiliations:** 1Institute of General Practice and Public Health, Claudiana—College of Health Professions, 39100 Bolzano, Italy; christian.wiedermann@am-mg.claudiana.bz.it; 2Department of Public Health, Medical Decision Making and HTA, University of Health Sciences, Medical Informatics and Technology—Tyrol, 6060 Hall, Austria

**Keywords:** human albumin, critical illness, meta-analysis, hemodynamic resuscitation, hypoalbuminemia

## Abstract

Background: Human albumin, a vital plasma protein with diverse molecular properties, has garnered interest for its therapeutic potential in various diseases, including critical illnesses. However, the efficacy of albumin infusion in critical care and its associated complications remains controversial. To address this, a review of recent meta-analyses was conducted to summarize the evidence pertaining to albumin use in critical illness. Methods: Adhering to the rapid review approach, designed to provide a concise synthesis of existing evidence within a short timeframe, relevant meta-analyses published in the last five years were identified and analyzed. PubMed, Embase, and Cochrane databases of systematic reviews were searched using pre-defined search terms. Eligible studies included meta-analyses examining the association between albumin infusion and outcomes in critically ill and perioperative patients. Results: Twelve meta-analyses were included in the review, covering diverse critical illnesses and perioperative scenarios such as sepsis, cardiothoracic surgery, and acute brain injury. The analyses revealed varying levels of evidence for the effects of albumin use on different outcomes, ranging from no significant associations to suggestive and convincing. Conclusions: Albumin infusion stabilizes hemodynamic resuscitation endpoints, improves diuretic resistance, and has the potential to prevent hypotensive episodes during mechanical ventilation in hypoalbuminemic patients and improve the survival of patients with septic shock. However, caution is warranted due to the methodological limitations of the included studies. Further high-quality research is needed to validate these findings and inform clinical decision-making regarding albumin use in critical care.

## 1. Introduction

Albumin is the most abundant plasma protein in humans and has long been recognized for its essential physiological functions in maintaining fluid balance, transporting molecules, and regulating pH. The therapeutic use of albumin is of significant interest because of its unique molecular characteristics, including its substantial antioxidant properties, versatile binding capabilities, and potential for targeted drug delivery [1]. Understanding the molecular basis of the therapeutic effects of albumin holds promise for advancing treatment strategies for various diseases and expanding the frontiers of molecular medicine.

Critical illness is a consequence of inflammation that causes progressive organ dysfunction [2]. It eventually enters a decompensated stage with life-threatening complications such as cardiovascular instability, respiratory failure, and renal dysfunction, which often results in high morbidity and mortality. Sepsis, trauma-related complications, burns, and perioperative states associated with major surgeries contribute to the development of critical illness [2]. In patients with critical illness, hypoalbuminemia is often observed due to changes in albumin kinetics related to capillary leakage, metabolism, and oxidative modifications of albumin molecules [3]. Hypoalbuminemia has been recognized as a significant predictor of poor outcomes [4].

The use of human albumin (HA) infusion has been extensively discussed for the management of critical illnesses and their associated complications [5]. HA infusion offers theoretical benefits owing to its oncotic and non-oncotic properties [6]. Oncotic properties help improve effective hypovolemia by expanding plasma volume, while non-oncotic properties influence various pathophysiological mechanisms such as binding and transport, detoxification, antioxidant effects, endothelial stabilization, and immunomodulation [7,8]. HA infusion has shown potential effectiveness in the treatment of complex liver cirrhosis [9]. However, the use of HA in critically ill non-liver patients, including those undergoing major surgery, remains controversial [10].

Meta-analyses play a crucial role in making informed clinical decisions as they combine all available data to provide the highest level of evidence [11]. However, a systematic assessment, including the quality of evidence and methodology in meta-analyses focusing on the efficacy of HA infusion for critical illness and its complications, is currently lacking. Therefore, the purpose of this rapid review is to summarize the evidence derived from meta-analyses pertaining to this subject.

## 2. Materials and Methods

This article is a compilation of existing studies and does not include any novel research involving human participants or animals. As a rapid review [12], it follows a streamlined methodology to provide a timely synthesis of evidence within a shorter timeframe. Notably, the study protocol has not been registered or published, which is common practice in systematic reviews. This review focuses on studies published within the last five years, with the aim of capturing the most recent evidence. Conducted by a single author, this review employs a narrow scope and excludes liver diseases. The literature search was limited to three databases: PubMed, Embase, and Cochrane Database of Systematic Reviews. The methodology used for quality assessment, data extraction, and synthesis was adapted to the expedited nature of the review process.

### 2.1. Search Strategy

PubMed, Embase, and Cochrane databases of scientific articles were searched for meta-analyses related to HA infusion from 1 January 2018 to 11 July 2023. The search was limited to studies published within the last five years, specifically from 2018 onwards. No language restrictions were applied. The identified articles were imported to Zotero, and duplicates were removed. In addition to the electronic search, manual identification of relevant meta-analyses was conducted using published reference lists. The search terms are listed in Appendix A.

### 2.2. Study Selection and Data Extraction

Meta-analyses of both RCTs and observational studies (cohort and case-control studies) investigating the association between HA use and mortality rates or other clinical and hemodynamic outcomes in critically ill perioperative patients were included. Articles containing multiple meta-analyses were eligible, and each meta-analysis was independently assessed for inclusion. The meta-analyses were identified based on the pre-defined eligibility criteria. Eligible studies should include all meta-analyses on HA infusion for the management of critically ill and perioperative patients. The exclusion criteria were as follows: (1) duplicated papers; (2) comments, guidelines, protocols, and editorials; (3) narrative reviews; (4) systematic reviews without meta-analyses; (5) network meta-analyses; (6) studies in which the target population was diagnosed with cirrhosis, not perioperative patients, or not critically ill; and (7) studies where HA treatment was not given. The publication of studies in the form of abstracts was not an exclusion criterion. There were no limitations on the publication language or date.

The following information was extracted from the included meta-analyses and recorded in a pre-designed table: first author, publication year, country, type and number of included studies, type and number of patients in the treatment and control groups, type of interventions employed in the treatment and control groups, objectives or indications of HA treatment, study heterogeneity, and random effects with a 95% confidence interval (CI).

Due to missing data, the authors of one of the studies [13] were contacted for the required information, and they provided supplemental data.

### 2.3. Methodological Quality Assessment

The methodological quality of the included meta-analyses was assessed using the Assessment of Multiple Systematic Reviews 2 (AMSTAR-2) tool [14]. This tool consists of 16 questions with “yes”, “partial yes”, and “no” answers. Of these, 7 questions (2, 4, 7, 9, 11, 13, and 15) were considered critical domains that had a significant impact on the validity and conclusions of the review. The overall confidence in the results of the meta-analyses was categorized as “high” (zero or one non-critical weakness), “moderate” (more than one non-critical weakness without critical weaknesses), “low” (one critical weakness), and “critically low” (more than one critical weakness). Only one investigator (CJW) conducted the assessment of methodological quality using the AMSTAR-2 tool [14].

### 2.4. Level of Evidence

The evidence level of the association between HA use and patient outcomes was determined for each meta-analysis. The criteria were based on statistical significance by random- or fixed-effects *p*-values, small-study effects, between-study heterogeneity, and concordance between the effect estimate of the largest study and the summary estimate of the meta-analysis. Heterogeneous data were used for descriptive purposes to highlight the variability and potential sources of inconsistency in the results.

The criteria were stratified as follows:Convincing evidence: (1) statistical significance at *p* < 0.001; (2) no small study effects or large between-study heterogeneity; and (3) concordance between the effect estimate of the largest study and the summary effect of the random-effects meta-analyses.Suggestive evidence: (1) statistical significance of random effects at *p* < 0.05; (2) 95% PI included the null hypothesis; and (3) no small study effects or large between-study heterogeneity.Weak evidence: (1) statistical significance of random effects at *p* < 0.05; (2) small study effects or large between-study heterogeneity were found.Non-significant association: There was no statistical significance for the random effect of the meta-analysis (*p* > 0.05).

## 3. Results

A comprehensive search of the PubMed, Embase, and Cochrane systematic review databases initially yielded 1706 papers. After screening and applying the inclusion and exclusion criteria, 12 meta-analyses were eligible for inclusion (Figure 1). The excluded studies are listed in Appendix A.

Table 1 presents the selection scheme for the inclusion of scientific articles and their respective characteristics. Of the 12 meta-analyses, 6 were conducted in Asia [15,16,17,18,19,20], 3 in North America [13,21,22], and 3 in Europe [23,24,25]. These studies cover a range of critically ill and perioperative conditions, including critical illness in hospital or emergency settings [13,20,23], sepsis [15,16], cardiothoracic surgery [17,21,24], and acute brain injury, such as ischemic stroke [19] and traumatic brain injury [25]. In some studies on mechanical ventilation [20] and diuretic resistance [18,22], the particular roles of hypoalbuminemia were investigated. For review purposes, each study was assessed based on its country of origin, patient population, number of trials, size, intervention compared to control, outcomes evaluated, results, and methodological quality according to the AMSTAR-2 tool [14].

The study sizes of the included meta-analyses varied significantly, reflecting the diversity of research conducted in different critical illness scenarios. The median number of trials in the 12 meta-analyses was 9.5 (interquartile range (IQR), 10.5). Detailed information on patient numbers was missing for two meta-analyses that were published only as abstracts [21,22]. Ten meta-analyses included the median number of patients with 1589 (IQR, 8184). 

### 3.1. Major Findings from Meta-Analyses

#### 3.1.1. Critical Illness

Two meta-analyses, Lewis et al. [23] and Martin and Bassett [13], investigated the use of HA in critically ill ICU patients. Both studies evaluated the use of colloids and crystalloids in critically ill patients. Lewis et al. [23] focused on the effects of different types of colloids compared with crystalloids on mortality, blood transfusion, and adverse events. Martin and Bassett [13] examined hemodynamic responses to crystalloids and colloids.

The review by Lewis et al. [23] included 22 RCTs with 13,047 participants. Different types of colloids (starches, dextrans, gelatins, and albumin/FFP) were compared with crystalloids. The analysis found that there was probably little or no difference in mortality between colloids and crystalloids. However, starches were associated with a slightly higher need for blood transfusion and renal replacement therapy than crystalloids were. The study concluded that the choice between colloids and crystalloids does not significantly impact mortality but may have implications for blood transfusion and renal replacement therapy with artificial colloids.

Martin and Bassett [13] identified 18 RCTs with 10,600 patients. The analysis found that central venous pressure (CVP) was significantly lower with crystalloids than with albumin, hydroxyethyl starch (HES), or gelatin. The mean arterial pressure (MAP) was also significantly lower with crystalloids than with albumin or gelatin. Crystalloids were administered in higher volumes than HES, and the cardiac index (CI) was lower in the crystalloid group than in the albumin group. The study concluded that crystalloids were less efficient than colloids in stabilizing hemodynamic resuscitation endpoints.

A third meta-analysis by Itagaki et al. [20] investigated the effects of HA and diuretics co-administration in mechanically ventilated patients with hypoalbuminemia. Patients treated with HA and diuretics experienced a reduced number of days on mechanical ventilation, fewer hypotensive events, and improved oxygenation at 24 h. However, in the relatively small number of 129 patients of the three studies included in the meta-analysis, there was no significant effect on all-cause mortality at 30 days.

#### 3.1.2. Sepsis

Two meta-analyses, Geng et al. [16] and Zou et al. [15], examined the use of HA in patients with sepsis. While both studies examined the effects of HA and crystalloids on mortality in sepsis and septic shock patients, Geng et al. [16] reported a potential benefit of HA, particularly at a 20% concentration, whereas Zou et al. [15] did not observe a significant reduction in mortality.

The results of Zou et al. [15] for patients with septic shock showed in the trial sequential analysis that the required information size was not achieved in all groups, suggesting that the effect size was not definitive.

Geng et al. [16] included eight studies with 5124 patients with sepsis and 3482 patients with septic shock. The analysis showed that compared to crystalloids, the use of HA may reduce the 90-day mortality in septic patients and significantly improve outcomes in septic shock patients. Specifically, the use of 20% HA significantly decreased 90-day mortality in patients with septic shock. Both 4–5% and 20% HA appeared to be more beneficial than crystalloids in improving the survival rate of patients with sepsis.

#### 3.1.3. Cardiac Surgery

Three meta-analyses conducted by Keshavarz et al. [21], Siemens et al. [24], and Wei et al. [17] focused on the use of HA in patients undergoing cardiothoracic surgery. None of the meta-analyses reported significant reductions in mortality among patients who received HA for cardiopulmonary bypass prime or intravenous infusion compared with those who did not receive HA.

The study by Keshavaz et al. [21] included 48 RCTs and found that intravenous HA did not significantly reduce mortality, kidney failure, length of hospital stay, length of ICU stay, or blood loss compared with no albumin. Wei et al. [17] compared the efficacy and safety of HA and 6% HES 130/0.4 in ten RCTs with a total of 1567 patients. The analysis showed no differences in mortality or total infusion between the two groups. However, given the safety concerns and the changing approval status of HES globally [26], the results of this study are likely to be secondary.

Siemens et al. [24] evaluated the effects of cardiopulmonary bypass (CPB) priming fluids on perioperative bleeding in pediatric cardiac surgery in 20 eligible RTCs involving 1550 patients. The study found no significant difference in postoperative bleeding between the different CPB priming solutions, including HA. The analysis was limited by the heterogeneity within the dataset, which compromised the ability to draw definitive conclusions.

#### 3.1.4. Diuretic Response

In addition to the study by Itagaki et al. [20] on the mechanical ventilation of hypoalbuminemic adults receiving diuretic therapy, two meta-analyses [18,22] explored the effectiveness of HA administration in clinical contexts characterized by diuretic resistance and low serum albumin levels, a condition commonly associated with critical illness and seen in fluid overload.

Lee et al. [18] explored the efficacy of the co-administration of furosemide and HA in achieving diuresis and natriuresis. The results indicated that co-administration increased urine output and sodium excretion compared with furosemide alone. The effectiveness of combination therapy is influenced by factors such as baseline serum albumin level, prescribed albumin infusion dose, and renal function. However, this study highlights the high heterogeneity in treatment response and calls for more parallel RCTs to provide further insights.

Chamarthi et al. [22] also focused on patients with hypoalbuminemia and diuretic resistance. The authors concluded that the co-administration of HA and furosemide increased urine output and sodium excretion at 8 h. However, there were no significant differences in urine volume at 24 h.

#### 3.1.5. Acute Brain Injury

Two studies analyzed the role of HA administration in two different neurological conditions: ischemic stroke and traumatic brain injury (TBI) [19,25].

In the case of ischemic stroke, a meta-analysis of four studies was conducted to evaluate the neurofunctional outcomes of patients treated with albumin therapy. The analysis revealed no statistically significant difference in the long-term neurological function between the albumin and control groups. Furthermore, there was concern about the occurrence of complications, such as pulmonary edema, after albumin infusion. Therefore, the administration of albumin therapy for acute ischemic stroke should be approached cautiously [19].

Patients with TBI often experience hypoalbuminemia and require fluid resuscitation. The use of HA for TBI is controversial [27], and the effects of different HA concentrations require further investigation. A meta-analysis of four controlled clinical studies focused on the use of hyperoncotic (20–25%) in TBI treatment [25]. The analysis showed that the intervention using hyperoncotic HA according to the Lund concept was associated with significantly reduced mortality compared with the control group. However, evidence of these beneficial effects is at a high risk of bias.

#### 3.1.6. Hypoalbuminemia, Burns, and Kidney Replacement Therapy

Hypoalbuminemia is frequently observed in hospitalized patients and is associated with several diseases [3]. In hypoalbuminemia, supplementation with HA affects surrogate outcome parameters, such as volume effects of resuscitation fluids or diuretic responses to furosemide [28,29]. In burn patients, HA may be administered acutely as a volume expander during burn shock resuscitation and chronically following resuscitation to correct hypoalbuminemia [30]. In hypoalbuminemic patients who need hemodialysis, HA administration before dialysis results in fewer episodes of hypotension and improves fluid removal [31]. However, in the past five years, no meta-analysis has been published on HA administration in patients with hypoalbuminemia, burns, or receiving kidney replacement therapy as principal inclusion criteria and main patient characteristics.

### 3.2. Methodological Quality of Meta-Analyses

The results of the assessments of the methodological quality of the meta-analyses are shown in Appendix A. Regarding the seven critical questions of the AMSTAR-2 tool, the following findings were observed among the included meta-analyses: three meta-analyses published a review protocol prior to conducting the review [18,20,23] (Q2); all of them performed a comprehensive literature search (Q4); three provided a list of excluded studies or reasons for exclusion [18,20,23] (Q7); seven meta-analyses employed a satisfactory approach for assessing the risk of bias [13,15,17,18,20,24] (Q9); all meta-analyses, except one published as congress report without showing details [22] used appropriate statistical methods for data synthesis (Q11); three meta-analyses failed to account for the risk of bias in individual studies when interpreting the results [16,21,22] (Q13); and seven meta-analyses thoroughly investigated publication bias [15,17,18,19,20,23,26] (Q15).

For the nine non-critical questions, the following findings were observed: all the included meta-analyses specified their inclusion and exclusion criteria, including the population, interventions, comparators, and outcomes (Q1), and explained their study design for inclusion criteria (Q3); one meta-analysis did not perform study selection in duplicate [25] and the two meta-analyses published as abstracts did not provide necessary information [21,22] (Q5); four meta-analyses did not conduct data extraction in duplicate [13,21,22,25] (Q6) and three failed to report their included studies in detail [19,21,22] (Q8); one meta-analysis reported the sources of funding for the studies included in the review [23] (Q10); three meta-analyses did not report or assess the potential impact of the risk of bias in each study [16,21,22] (Q12); eight meta-analyses explored possible reasons for heterogeneity and discussed its effect on the results [13,15,16,18,19,20,23,25] (Q14); and except for the two abstract reports [21,22] all meta-analyses claimed no conflict of interest (Q16).

In general, the methodological quality varied among the studies, with some being rated as high quality while others were considered critically low quality. Of the assessed meta-analyses, only two studies rated ‘high quality’ [18,23]:A meta-analysis by Lewis et al. [23] combined HA and fresh frozen plasma as natural colloids despite their significant pharmacological differences, which may confound the interpretation of the results, indicating that the choice between colloids and crystalloids does not significantly impact mortality.A meta-analysis by Lee et al. [18] from Taiwan focused on patients receiving diuretic therapy. The effectiveness of combination therapy was influenced by factors such as baseline serum albumin levels, prescribed HA infusion doses, and renal function.

The majority of the included meta-analyses were AMSTAR-2, rated as ‘critically low’ [13,16,17,19,24,25], of which two meta-analyses were presented at scientific meetings and published only as abstracts [21,22].

Two studies exhibited intermediate quality and were rated ‘low’ [15,20].

Itagaki et al. [20] from Japan focused on critically ill patients with hypoalbuminemia who received mechanical ventilation. As the effects of HA were analyzed in combination with diuretics, these findings may supplement the results of a high-quality study by Lee et al. [18].In the second ‘low-quality’ meta-analysis by Zou et al. [15], the use of HA was compared with crystalloids in a larger number of patients with septic shock. As described, the analysis did not find a decrease in all-cause mortality in patients treated with HA compared to those treated with crystalloids.

### 3.3. Level of Evidence of Human Albumin Associations with Outcomes

Table 2 presents the level of evidence of the association between HA use and various outcomes in critically ill and perioperative patients. All studies were based on RCTs, except for one meta-analysis on TBI in four clinical trials that included one RCT and three observational studies [25].

‘Convincing evidence’ (effect size at *p* < 0.001, absence of heterogeneity between included trials, no small study effect, and concordance between the effect estimate of the largest study and the summary effect of the random-effects meta-analyses) for an association of HA use with outcome versus crystalloids was found for CI as hemodynamic resuscitation endpoint in critically ill patients [13]. This finding is supported by ‘weak evidence’ (*p* < 0.05, significant heterogeneity *p* < 0.05) for increases in MAP and CVP with HA administration [13]. Prevention of hypotensive episodes during mechanical ventilation in hypoalbuminemic patients, observed by Itagaki et al. [20], may also be related to HA’s significant association with improved hemodynamics as indicated by ‘suggestive evidence’ (effect size *p* < 0.05, no heterogeneity *p* > 0.05) for this outcome. The methodological quality of the study by Itagaki et al. [20] was rated as ‘low’; the quality of the hemodynamic endpoint study by Martin and Bassett [13], however, was assessed as ‘critically low’ (Appendix A).

In addition to the prevention of hypotensive episodes in hypoalbuminemic patients during mechanical ventilation [20], ‘suggestive evidence’ for the beneficial outcome effects of HA also exists for mortality in patients with septic shock [16] and after severe traumatic brain injury when intensive care is provided according to the Lund concept using 20–25% HA [25]. However, the clinical significance of these two results is limited not only by the suggestive level of evidence but also by the ‘critically low’ rating of the methodological quality of the meta-analyses (Appendix A). This is especially true for the study on TBI because a high risk of bias in this meta-analysis was highlighted in the original publication, including the use of observational studies, limited study sizes, and differences in concomitant therapies as comparator groups of HA [25].

At the ‘weak evidence’ level, two meta-analyses on the effect of HA on urine production and sodium excretion consistently described, in different types of patients, that the administration of HA leads to a transient increase in the efficacy of diuretics [18,22]. One of the studies was of methodological ‘high quality’ [18], and the other ‘critically low’ [22].

In several meta-analyses and outcomes, ‘no significant association’ between HA use and outcomes was observed, including mortality in critical illness [13,20,23], sepsis [16], septic shock [15], and surgery [17,21]; AKI, kidney failure, or KRT in critical illness [23] and surgery [17,21]; LOS in ICU and hospital [17,21] and blood loss [21,24] in surgery; transfusion in critical illness [23] and surgery [17]; infusion volume during surgery [17]; and neurological outcome in ischemic stroke [19]. Most studies have ‘critically low’ methodological quality [13,16,17,19,21,24].

Overall, the meta-analyses provided varying levels of evidence for the effects of HA use in critically ill and perioperative patients, ranging from ‘no significant associations’ between HA use and the investigated outcomes, ‘weak evidence’, to ‘suggestive evidence’, and in the case of hemodynamic resuscitation endpoints, ‘convincing evidence’. However, these findings must be interpreted cautiously, considering the methodological quality and potential biases of the included studies. Further high-quality research is required to confirm these findings.

## 4. Discussion

The utilization of HA has experienced variations over time, partly due to varying and sometimes conflicting conclusions drawn from clinical studies, as well as the absence of clear guidelines and prevailing misconceptions [33].

This study aimed to provide a comprehensive overview of the major findings of recent meta-analyses examining the use of HA in critically ill and perioperative patients. The results revealed heterogeneous effects of HA administration on different clinical outcomes, emphasizing the importance of methodological quality in interpreting evidence. These findings shed light on the effects of HA on critical illness, sepsis, cardiac surgery, and diuretic response. In critically ill patients, the choice between colloids, including HA and crystalloids, did not significantly impact mortality, although crystalloids were less efficient than HA in stabilizing resuscitation endpoints. HA administration showed potential benefits, particularly in reducing mortality, in patients with septic shock, although some studies did not observe a significant effect. In cardiac surgery, HA did not result in a significant reduction in mortality or other important outcomes. When combined with diuretics, HA was found to enhance diuretic and natriuretic effects, but there was high heterogeneity in the treatment response. In the case of ischemic stroke, HA did not beneficially affect long-term neurological function, and concerns have been raised regarding cardiac complications after albumin infusion. For traumatic brain injury, the use of 20–25% HA according to the Lund concept was associated with significantly reduced mortality, but the evidence is of critically low quality. Notably, no recent meta-analyses have investigated the use of HA in patients with hypoalbuminemia aiming for substitution, burns, or kidney replacement therapy.

Colloids are commonly used in large-volume fluid resuscitation [34]. The ‘Saline versus Albumin Fluid Evaluation’ (SAFE) study, which investigated hypotonic 4% HA versus saline 0.9% in critically ill patients, sheds light on the volume effect of HA in comparison to other resuscitation fluids. It was found that the ratio of administered HA to saline volume required to achieve hemodynamic targets in the first four days was approximately 1 to 1.4 [35]. The stronger volume effect of HA was even more pronounced in patients with hypoalbuminemia [29]. In addition, in the ‘Volume Replacement With Albumin in Severe Sepsis’ (ALBIOS) study, which compared the administration of 20% HA, dosed by albumin serum levels, in sepsis patients with 0.9% NaCl, it was confirmed that during the first seven days, patients in the HA group, as compared with those in the crystalloid group, had a higher MAP and lower net fluid balance [36]. In addition, both large, high-quality RCT studies confirmed that HA administration is safe, and relevant side effects such as renal impairment, bleeding, prolonged LOS, or an increase in mortality were not observed [35,36]. These observations form an important basis for the recommendation in the Surviving Sepsis Campaign Guidelines to use HA in volume resuscitation of patients with sepsis when crystalloids alone are insufficient [37].

Findings from this rapid review align with previous safety observations made in landmark studies, such as SAFE and hemodynamic superiority observed in ALBIOS. The meta-analysis by Martin and Bassett [13] provides convincing evidence of HA’s association with improved hemodynamic outcomes in a heterogeneous group of patients and is supported by the reduction in hypotensive episodes with the use of HA in hypoalbuminemic patients during mechanical ventilation when diuretics are needed [20].

While Lewis et al. [23] focused on the effects of different types of colloids compared with crystalloids on mortality, blood transfusion, and adverse events, Martin and Bassett [13] examined the hemodynamic response to crystalloids and colloids. Neither study in critically ill patients found any significant association between HA use and mortality outcomes. Notably, these meta-analyses were rated as ‘high’ and ‘critically low’ methodological quality, respectively. Nonetheless, due consideration must be given to the evolving landscape of evidence and the complexity of individual patient contexts when determining the appropriate use of HA in critical care settings.

For patients with septic shock, two meta-analyses by Geng et al. and Zou et al. explored the effects of HA compared with crystalloids on mortality [12,13]. Geng et al. reported a potential benefit of HA, particularly at 20% concentration, in reducing mortality, whereas Zou et al. did not observe a significant reduction in mortality. The results for septic shock patients in the study by Zou et al. were limited by the small sample size, indicating the need for further investigation. Both meta-analyses were rated as ‘critically low’ methodological quality. The ‘Efficacy of Albumin Replacement and Balanced Solution in Patients With Septic Shock Trial’ (the ALBIOSS-BALANCED Trial), a 2-by-2 factorial, investigator-initiated, open-label, multicenter RCT measuring the effects of 20% HA and balanced crystalloids versus crystalloids alone on all-cause death from randomization to 90 days and the composite of all-cause death from randomization to 90 days and new occurrence of AKI as coprimary endpoints [38] which is about to be completed, may successfully provide answers to open questions about the effects of HA on the survival of patients with sepsis [33] and better inform clinical practice.

In the context of cardiac surgery, three meta-analyses by Keshavarz et al., Siemens et al., and Wei et al. examined the use of HA in patients who underwent cardiothoracic surgery [14,19,22]. None of these studies reported significant reductions in mortality among patients who received HA compared to those who did not, and the association of HA use with other outcomes is hardly evaluable as the methodological quality of these meta-analyses was rated as ‘critically low’: one study was reported only as an abstract [21], another one used an artificial colloid as comparator fluid, i.e., HES being taken off the market [17], and the third meta-analysis covered pediatric cardiac surgery [24]. Wei et al. [17] reported evidence of increased blood loss in 216 patients receiving HA compared to 222 patients receiving HES during cardiac surgery, without an increase in transfusion requirements. Increased bleeding in the HA group compared with the HES group in adult cardiac surgery contradicts previous meta-analytic evidence [39] and may be prone to bias [26]. The use of HES has recently been severely restricted owing to safety concerns [32]. In addition, there is strong evidence of reporting bias in the meta-analysis [26], further weakening the evidence. Thus, meta-analyses of the last five years do not add to the body of knowledge on the use of HA in cardiac surgery.

Recent data from an RCT not yet included in the meta-analyses showed that perioperative administration of 4% HA resulted in increased blood loss compared to Ringer’s acetate; the effect was comparable in magnitude to the complexity and urgency of the procedure [40]. Previously, colloids interfered with blood coagulation and produced greater hemodilution, which was associated with more transfusions of blood products compared with crystalloid use only [41]. Thus, bleeding complications with 4% HA compared with crystalloids in cardiac surgery are a potential safety concern that requires further clarification, including studies of small-volume resuscitation with 20–25% HA [42].

Two meta-analyses by Lee et al. [18] and Chamarthi et al. [22] investigated the effectiveness of HA administration in patients with diuretic resistance and hypoalbuminemia. Both studies found that the co-administration of HA and diuretics resulted in increased urine output and sodium excretion compared to diuretics alone. One of the two meta-analyses rated the methodological quality [18]. The role of HA in diuretic resistance is significant, as severe hypoalbuminemia can contribute to this condition through various mechanisms. Reduced delivery of diuretics to tubules occurs because furosemide binds to albumin, which requires renal blood flow to reach the proximal tubule. Additionally, reduced intravascular volume may limit the fluid available for removal, and in patients with proteinuria, frusemide may bind to albumin in the intratubular space [43]. Further high-quality research is required to confirm and better understand the potential clinical benefits of this combination therapy. Additionally, investigating whether and how overcoming diuretic resistance in the acute care of critically ill patients affects their clinical outcomes remains important.

For acute brain injury, meta-analyses on ischemic stroke and TBI have been conducted by different authors [16,23]. A study on ischemic stroke did not find any significant difference in long-term neurological function between the HA and control groups [19]. In contrast, the TBI meta-analysis suggested a significant reduction in mortality with hyperoncotic HA according to the Lund concept.

The potential impact of HA on mortality in TBI remains uncertain, and the current evidence presents mixed findings. In the ‘Balanced Solution Versus Saline in Intensive Care Study’ (BaSICS), where hypotonic balanced crystalloids were compared with 0.9% saline, TBI patients treated with saline showed significantly better 90-day survival [44]. In contrast, the SAFE-TBI study, a post-hoc follow-up analysis of 460 patients from the SAFE trial [45], reported higher mortality in those who received 4% HA compared to 0.9% saline. It is worth noting that the low osmolality of HA used (266–267 mOsmol/L H_2_O) in the SAFE-TBI study might have been suboptimal for patients with TBI. Experimental studies have directly compared commercially available hypotonic 4% HA used in SAFE with isotonic 4% HA (theoretical osmolarity, 288 mOsmol/kg) and revealed a higher ICP with hypotonic HA, suggesting that tonicity rather than albumin itself may impact ICP [46]. Thus, the current evidence on the association between HA use and mortality in TBI is inconclusive. Further research is needed to elucidate the specific impacts of different types of HA solutions and their effects on patient outcomes in the context of TBI.

No recent meta-analysis has specifically addressed hypoalbuminemia, KRT, or burns. Previous studies provided important evidence regarding these topics. These findings suggest that hypoalbuminemia is associated with worse outcomes, and the administration of HA in certain clinical contexts, such as during KRT, may positively impact fluid removal and prevent hypotension [33]. Nonetheless, further research is required to gain a comprehensive understanding of these aspects and to explore potential new findings in these critical care settings.

Despite the comprehensive analysis provided in this study, it is essential to interpret the findings cautiously because of the varying methodological qualities of the included studies. This narrative rapid review has inherent limitations to consider. Firstly, its focus on the last five years of literature may exclude earlier valuable insights into albumin use in critical care, prioritizing recent developments but limiting historical context. Additionally, its non-systematic search, differing from PRISMA guidelines, may lead to omissions of relevant meta-analyses and systematic reviews, potentially introducing selection bias. Furthermore, relying on a single author for article selection could introduce subjectivity. Another limitation of this review is that it relied on the use of the term ‘meta-analysis’ as a filter in PubMed’s search criteria. While efforts were made to capture all relevant meta-analyses, it is acknowledged that the process of systematization by PubMed may result in some meta-analyses not being promptly marked as such, potentially leading to the omission of relevant studies. Lastly, the absence of PROSPERO registration means the review process, including search strategy and inclusion criteria, lacks pre-specification and public documentation. These limitations underscore the need for caution in interpreting these findings. The goal of this work is to offer a valuable synthesis of evidence within these constraints, recognizing that further research and systematic reviews may provide additional insights. Further high-quality research is warranted to confirm the association between HA use and the investigated outcomes in critically ill and perioperative patients. Clinicians should carefully consider the available evidence when making decisions regarding HA use in critically ill and perioperative patients.

## 5. Conclusions

This rapid review presents a comprehensive analysis of recent meta-analyses that explored the use of HA in critically ill and perioperative patients. These findings highlight the heterogeneous effects of HA administration on various clinical outcomes and underscore the importance of considering methodological quality when interpreting evidence. While no significant impact on mortality was observed in critically ill patients, HA demonstrated potential benefits in reducing mortality in patients with septic shock. However, evidence from the context of cardiac surgery remains inconclusive. Combining HA with diuretics enhances diuresis and natriuresis, particularly in patients with diuretic resistance and hypoalbuminemia. Notably, no recent meta-analyses have investigated HA use in patients with hypoalbuminemia aiming for substitution, burns, or KRT. The utilization of HA in large-volume fluid resuscitation is supported by landmark studies, such as SAFE and ALBIOS, which documented its safety and hemodynamic superiority. While evidence suggests improved hemodynamic outcomes with HA, such as in hypoalbuminemic patients during mechanical ventilation with diuretics, further research is needed to confirm these findings. Moreover, the impact of HA on mortality in patients with TBI remains uncertain, with conflicting evidence. Therefore, clinicians should cautiously consider the available evidence when making decisions regarding the use of HA in critically ill and perioperative patients, recognizing the need for further high-quality research to better inform clinical practice in these settings.

## Figures and Tables

**Figure 1 jcm-12-05919-f001:**
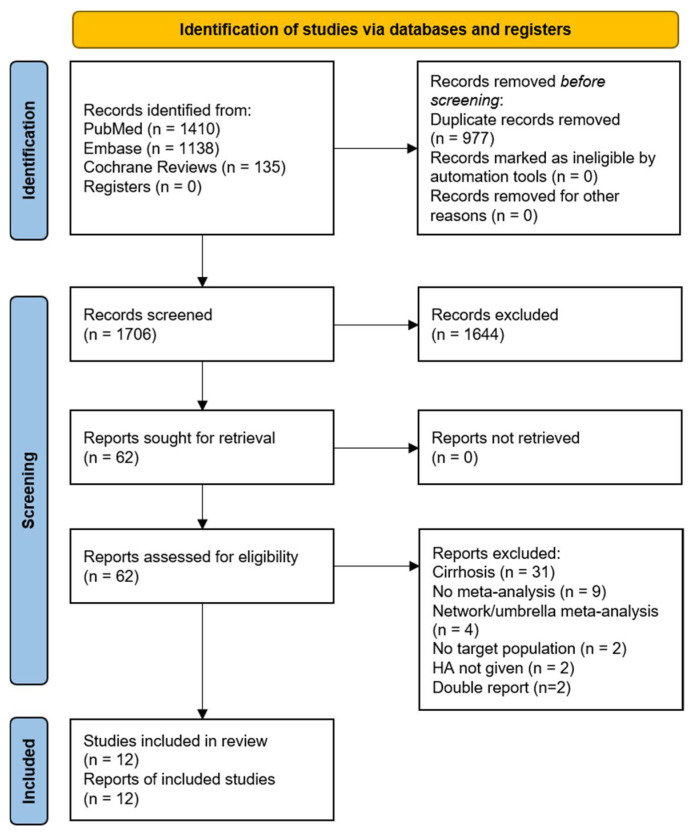
PRISMA 2020 flow diagram for new systematic reviews, which included searches of databases and registers only. HA, human albumin.

**Table 1 jcm-12-05919-t001:** Characteristics and findings of included meta-analyses.

Study	Year	Country	Population	Trials	Patients	Intervention vs. Comparison	Outcomes	Results	Methodological Quality *
Critical Illness									
Lewis et al. [23]	2018	United Kingdom	Critically ill receiving fluid volume replacement in hospital or emergency out-of-hospital settings	22	13,047	Natural colloids (HA and FFP) vs. crystalloid (isotonic or hypertonic)	Mortality, need for blood transfusion, need for KRT	Little or no difference in mortality, little or no difference in need for KRT, uncertain evidence for blood transfusions.	High
Martin and Bassett [13]	2019	United States	Critically ill receiving fluid resuscitation in intensive care units	18	10,600	HA vs. crystalloid	Hemodynamics (MAP, CVP, CI) at end of fluid challenge), mortality	HA more efficient than crystalloids at stabilizing resuscitation endpoints.	Critically low
Itagaki et al. [20]	2022	Japan	Critically ill with hypoalbuminemia receiving mechanical ventilation	3	129	HA with diuretics vs. placebo or no diuretics	Hypotensive events, duration of mechanical ventilation, all-cause mortality	Reduced number of days on mechanical ventilation, reduced hypotensive events, and improved P/F ratio at 24 h, no effect on all-cause mortality at 30 days.	Low
Sepsis									
Zou et al. [15]	2018	China	Septic shock	6	3088	HA vs. crystalloid	Mortality	No decrease in all-cause mortality, according to trial sequential analysis, could be false negative.	Low
Geng et al. [16]	2023	China	Severe sepsis or septic shock	8	8606	HA vs. crystalloid	Mortality	Trend toward reduced 90-day mortality of septic patients, significantly reduced mortality of septic shock patients, particularly with 20% HA.	Critically low
Cardiothoracic surgery									
Keshavarz et al. [21]	2021	Canada	Adult cardiothoracic surgery	48	n.d.	HA vs. no HA or alternative solutions for infusion or bypass prime	Mortality, kidney failure, hospital LOS, ICU LOS, blood loss	No effects on mortality, kidney failure, hospital LOS, ICU LOS, and blood loss.	Critically low
Siemens et al. [24]	2022	United Kingdom	Pediatric cardiac surgery	12	881	HA vs. FFP, HES, crystalloids, gelatin, or HA (lower concentration) as alternative solution for cardiopulmonary bypass prime	Blood loss, transfusion requirement, coagulation parameters, ventilation duration, ICU LOS, hospital LOS, mortality	No differences in blood loss (other outcomes not assessed because of heterogeneity risk of bias).	Critically low
Wei et al. [17]	2021	China	Adult cardiac surgery	10	1567	HA vs. 6% HES 130/0.4	Mortality, AKI, KRT, hospital LOS, ICU LOS, blood loss, transfusion requirement, volume of infusion	Higher blood loss in HA group, no differences in transfusion requirement or other outcomes.	Critically low
Diuretic Resistance									
Chamarthi et al. [22]	2019	United States	Diuretic therapy in hypoalbuminemia	9	n.d.	HA plus diuretic vs. diuretic alone	Urine output, sodium excretion	Increased urine volume in HA group at 8 h, no difference after 24 h.	Critically low
Lee et al. [18]	2021	Taiwan	Diuretic therapy	13	422	HA plus diuretic vs. diuretic alone	Urine output, sodium excretion	Enhanced diuresis and natriuresis with advantages with baseline serum albumin levels lower than 2.5 g/dL, higher HA doses, and impaired renal function.	High
Acute Brain Injury									
Huang et al. [19]	2021	China	Ischemic stroke	4	1611	HA during the acute phase vs. no HA	Neurological outcome	No beneficial effect on the long-term neurological function, risk of pulmonary edema.	Critically low
Wiedermann [25]	2022	Italy	Severe TBI	4	320	ICP-targeted HA vs. CPP-targeted or standard therapy with no HA	Mortality	Reduced mortality but high risk of bias.	Critically low

* Methodological quality according to the Assessment of Multiple Systematic Reviews 2 (AMSTAR-2) tool [14]. AKI, acute kidney injury; CI, cardiac index; CPP, cerebral perfusion pressure; CVP, central venous pressure; FFP, fresh frozen plasma; HA, human albumin; HES, hydroxyethyl starch; ICP, intracranial pressure; KRT, kidney replacement therapy; LOS, length of stay; MAP, mean arterial pressure; n.d., no data; P/F, ratio of arterial partial pressure of oxygen (PaO_2_) to the fraction of inspired oxygen (FiO_2_); TBI, traumatic brain injury.

**Table 2 jcm-12-05919-t002:** Association of albumin use and outcomes in critically ill and perioperative patients.

Type of Patients	Outcome	Study	Type of Study	Studies	Patients (Intervention/Comparator)	Type of Metrics	Effect Size (95% CI)	Effect Size *p*–Value	Heterogeneity *p*–Value (I^2^)	Egger	Small Study Effect	Effect Size Concordance with Largest Study
Non-significant association (*p* > 0.05)												
Critically ill	Mortality	Lewis et al. [23]	RCT	20	6021/7026	random	0.98 (0.92, 1.06) ^$^	0.63	0.38 (6.61)	n.d.	no	n.a.
		Martin and Bassett [13]	RCT	14	5240/5240	random	1.02 (0.96, 1.10) ^&^	0.49	0.60 (0)	0.25	no	n.a.
	Renal (kidney failure)	Lewis et al. [23]	RCT	2	1506/1522	random	1.11 (0.96, 1.27) ^$^	0.15	0.45 (0)	n.d.	n.d.	n.a.
	Transfusion	Lewis et al. [23]	RCT	3	130/160	random	1.31 (0.95; 1.80) ^&^	0.10	0.56 (0)	n.d.	n.d.	n.a.
	Infusion Volume	Martin and Bassett [13]	RCT	5	555/566	random	1985 (−401, 4372) ^‡^	0.10	<0.001 (94)	0.31	no	n.a.
Septic shock	Mortality	Zou et al. [15]	RCT	6	1282/1806	random	0.91 (0.83, 1.00)	0.05	0.63 (0)	n.d.	n.d.	n.a.
Sepsis	Mortality	Geng et al. [16]	RCT	8	2302/2808	fixed	0.91 (0.80, 1.02) ^&^	0.11	0.64 (0)	n.d.	n.d.	n.a.
Surgery	Mortality	Keshavarz et al. [21]	RCT	8	606/617	random	0.01 (−0.01, 0.02) ^†^	n.d.	n.d.	n.d.	n.d.	n.a.
		Wei et al. [17]	RCT	3	163/171	fixed	0.56 (0.16, 2.02) ^&^	0.46	0.41 (0)	n.d.	n.d.	n.a.
	Renal (kidney failure)	Keshavarz et al. [21]	RCT	3	169/170	random	−0.03 (−0.22, 0.15) ^†^	n.d.	n.d.	n.d.	n.d.	n.a.
	Renal (AKI)	Wei et al. [17]	RCT	2	90/87	random	1.25 (1.00, 1.75) ^$^	0.05	0.99 (0)	n.d.	n.d.	n.a.
	Renal (KRT)	Wei et al. [17]	RCT	2	148/150	random	0.67 (0.08, 5.75) ^$^	0.72	0.18 (44)	n.d.	n.d.	n.a.
	LOS (hospital)	Keshavarz et al. [21]	RCT	9	510/529	random	−0.04 (−0.20, 0.12) ^‡^	n.d.	n.d.	n.d.	n.d.	n.a.
		Wei et al. [17]	RCT	4	172/176	fixed	−0.11 (−0.32, 0.10) ^‡^	0.32	0.33 (13)	n.d.	n.d.	n.a.
	LOS (ICU)	Keshavarz et al. [21]	RCT	4	250/259	random	0.02 (−0.20, 0.24) ^‡^	n.d.	n.d.	n.d.	n.d.	n.a.
		Wei et al. [17]	RCT	6	235/243	fixed	−0.18 (−0.36, 0.00) ^‡,§^	0.05 ^§^	0.59 (0)	n.d.	n.d.	n.a.
	Blood loss	Keshavarz et al. [21]	RCT	10	515/543	random	−0.16 (−0.34, 0.01) ^‡^	n.d.	n.d.	n.d.	n.d.	n.a.
		Siemens et al. [24]	RCT	2	100/116	random	4.51 (−1.18,10.19)	0.12	0.03 (80)	n.d.	n.d.	n.a.
	Transfusion	Wei et al. [17]	RCT	7	287/292	fixed	1.11 (0.95, 1.27) ^$^	0.20	0.19 (31)	n.d.	n.d.	n.a.
	Infusion volume	Wei et al. [17]	RCT	7	284/295	fixed	0.04 (−0.12, 0.20) ^‡^	0.64	0.50 (0)	n.d.	n.d.	n.a.
Mechanical ventilation (hypoalbuminemia)	Mortality	Itagaki et al. [20]	RCT	3	66/63	random	1.00 (0.45; 1.23) ^$^	1.00	0.30 (17)	n.d.	n.d.	n.a.
	Duration of ventilation	Itagaki et al. [20]	RCT	2	46/43	fixed	−0.34 (1.00; 1.31) ^$^	0.69	0.45 (0)	n.d.	n.d.	n.a.
Ischemic stroke	Neurological outcome	Huang et al. [19]	RCT	4	807/804	fixed	1.04 (0,85; 1.27) ^&^	0.72	0.65 (0)	n.d.	n.d.	n.a.
Diuretic resistance	Urine output (ml increase at 24 h)	Chamarthi et al. [22]	n.d.	9	n.d.	n.d.	385 (−141.92, 911.68) ^‡^	n.d.	n.d.	n.d.	n.d.	n.a.
Weak evidence (*p* < 0.05, significant heterogeneity *p* < 0.05)												
Critically ill	Hemodynamics (MAP max)	Martin and Bassett [13]	RCT	6	5175/3774	random	−3.5 (−6.71, −0.36) ^‡^	<0.001	<0.001 (91)	0.21	no	n.a.
	Hemodynamics (CVP max)	Martin and Bassett [13]	RCT	7	5187/3786	random	−2.0 (−3.0, −1.1) ^‡^	<0.001	<0.001 (82)	0.10	no	n.a.
Diuretic resistance	Urine output (ml increase per hour)	Lee et al. [18]	RCT	14	223/220	random	31.45 (19.30, 43.59)	n.d.	<0.01 (87)	n.d.	no	n.a.
	Urine output (ml increase at 8 h)	Chamarthi et al. [22]	n.d.	9	n.d.	n.d.	315 (183.04, 448.33) ^‡^	n.d.	n.d.	n.d.	n.d.	n.a.
	Sodium excretion (meq increase, 8 h)	Chamarthi et al. [22]	n.d.	9	n.d.	n.d.	27 (7.46, 46.59) ^‡^	n.d.	n.d.	n.d.	n.d.	n.a.
	Sodium excretion (meq increase per hour)	Lee et al. [18]	RCT	11	233/233	random	1.76 (0.83, 2.69)	n.d.	<0.01 (92)	n.d.	no	n.t.
Suggestive evidence (effect size *p* < 0.05, I^2^ *p*-value > 0.05)												
Septic shock	Mortality	Geng et al. [16]	RCT	7	1488/1994	fixed	0.85 (0.74, 0.99) ^&^	0.04	0.60 (0)	n.d.	n.d.	yes
Severe TBI	Mortality	Wiedermann [25]	OBS/RCT	1/3	165/155	random	0.42 (0.24; 0.73) ^$^	0.002	0.18 (38.6)	0.65	n.d.	yes
Surgery	Blood loss	Wei et al. [17]	RCT	6	216/222	fixed	0.22 (0.03, 0.41) ^‡^	0.02	0.60 (0)	n.d.	no	yes
Mechanical ventilation (hypoalbuminemia)	Hypotensive events	Itagaki et al. [20]	RCT	3	66/63	random	0.34 (0.15; 0.81) ^$^	0.01	0.91 (0)	n.d.	n.d.	yes
Convincing evidence (*p* < 0.001, no heterogeneity, concordance between the effect estimate of the largest study and the summary effect of the random-effects meta-analyses)												
Critically ill	Cardiac index (end of fluid challenge)	Martin and Bassett [13]	RCT	7	130/132	random	−0.61 (−0.87, −0.34) ”	<0.001	0.36 (15)	0.65	no	yes

^†^ Risk difference, ^‡^ standardized mean difference, ^$^ risk ratio, ^&^ odds ratio, ”calculated as values in the crystalloid group relative to the colloid group. ^§^ Re-calculation of statistics at higher precision showed that ICU length of stay of patients in the human albumin group was significantly shorter than that of patients in the 6% hydroxyethyl starches 130/0.4 group (standard mean difference −0.181, 95% confidence interval −0.361 to −0.001, *p* = 0.049) [32]. CPP, cerebral perfusion pressure; FFP, fresh frozen plasma; HA, human albumin; ICP, intracranial pressure; KRT, kidney replacement therapy; n.a., not assessed; n.d., not documented; OBS, observational study; TBI, traumatic brain injury.

## Data Availability

No new data were created.

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
