# Peer review of "Human Albumin Infusion in Critically Ill and Perioperative Patients: Narrative Rapid Review of Meta-Analyses from the Last Five Years"

_jcm, 2023, doi:10.3390/jcm12185919_

Round 1

Reviewer 1 Report

The manuscript presents a rapid Review of meta-analyses from the last five years on human albumin infusion in critically ill and perioperative patients. The research is well-conducted, and the writing is clear and coherent. The study design, methodology, and data analysis are commendable. A minor revision is necessary to further enhance the manuscript's clarity before it can be recommended for publication.

1. Thorough Literature Review: The manuscript provides a comprehensive review of current literature on human albumin infusion in critically ill and perioperative patients. This contextualizes the study's objectives and highlights its relevance within the field.

2. Methodology and Data Collection: The research design is robust, and the methods used for data collection are clearly described. The inclusion criteria for participants and the randomization process are well-elaborated.

3. Data Analysis: The statistical analysis is appropriate and effectively supports the study's conclusions. The presentation of results through tables is clear and aids in understanding.

4. Discussion: The discussion section effectively interprets the findings in the context of existing knowledge. The implications of the study's results for clinical practice and future research are well-explored.

Language and Style:

The manuscript is well-written with appropriate English language usage. The text flows smoothly, and the scientific terminology is accurately used. The manuscript demonstrates a high level of readability.

Minor Revisions:

1. Clarity in Methodology Description:

“In addition, one investigator (CJW) assessed  methodological quality.” Line 112 - 113

Based on which criteria? Objective criteria? Please explain.

Kind regards

Author Response

Point-by-point Response to Reviewer 1

COMMENT

RESPONSE

The manuscript presents a rapid Review of meta-analyses from the last five years on human albumin infusion in critically ill and perioperative patients. The research is well-conducted, and the writing is clear and coherent. The study design, methodology, and data analysis are commendable. A minor revision is necessary to further enhance the manuscript's clarity before it can be recommended for publication.

Thank you very much for your positive feedback and for recognizing the quality of research and manuscript. I greatly appreciate your valuable input.

1. Thorough Literature Review: The manuscript provides a comprehensive review of current literature on human albumin infusion in critically ill and perioperative patients. This contextualizes the study's objectives and highlights its relevance within the field.

Thank you for acknowledging the thoroughness of our literature review.

2. Methodology and Data Collection: The research design is robust, and the methods used for data collection are clearly described. The inclusion criteria for participants and the randomization process are well-elaborated.

Thank you for acknowledging that you found our methodology and data collection processes to be well-described and robust.

3. Data Analysis: The statistical analysis is appropriate and effectively supports the study's conclusions. The presentation of results through tables is clear and aids in understanding.

We are pleased that you also found the analyses and presentation of results appropriate and clear.

4. Discussion: The discussion section effectively interprets the findings in the context of existing knowledge. The implications of the study's results for clinical practice and future research are well-explored.

I appreciate your feedback regarding the discussion section of our manuscript. My aim was indeed to provide a comprehensive interpretation of the findings within the context of existing knowledge and to explore the implications of our results for clinical practice and future research.

Language and Style:

The manuscript is well-written with appropriate English language usage. The text flows smoothly, and the scientific terminology is accurately used. The manuscript demonstrates a high level of readability.

Thank you again for your appreciation.

Minor Revisions:

1. Clarity in Methodology Description: “In addition, one investigator (CJW) assessed methodological quality.” Line 112 – 113. Based on which criteria? Objective criteria? Please explain.

To improve the clarity, the former lines 112-113 have been changed to

Only one investigator (CJW) conducted the assessment of methodological quality using the AMSTAR-2 tool [14].

Reviewer 2 Report

Thank you so much for providing me with the opportunity to review the manuscript. 

The author has done an phenomenol job of reviewing recent literature on use of albumin in critically ill and peri-operative patients.

I honestly, do no much to add to the manuscript as it succint and to the point.

Author Response

Thank you for your kind words and valuable feedback. Your positive assessment is greatly appreciated.

Reviewer 3 Report

The author summarized finding of recent meta-analyses about utilization of human serum albumin infusion in critically ill and perioperative patients.

I recommend that this very well written manuscript describing scientifically important theme could be published after only slight minor corrections.

The main strength of the manuscript is strict and appropriate methodological approach that is used during literature search, selection of articles and analysis and interpretation of obtained data. 

There are several points for suggested minor revision:

1. The modification of title could be considerate to avoid repeating of term review; the readers could become confused if it is narrative review of rapid review; My suggestion for title may be Effectiveness of Clinical Use of Human Albumin in Critically Ill and Perioperative Patients: Rapid Review of Recent Meta-Analyses 

3. Line 36 It may be useful to add that albumin play significant antioxidant roles

Line 37 I suggest You to change term molecular science to clinical medicine molecular science

Line 57 Controversy should be changed to controversial

Line 76. It may be more appropriate to change systematic reviews term with scientific articles as this databases include other types of paper beside systematic reviews too

Section 2.2. I suggest You to rewrite sentences line88-89, line 97-102 in passive voice to avoid using the initials of the authors in the text; the contribution of the author/s is/are usually described only in the paragraph that at the end of the text 

Line 142 The included studies could be changed to selection scheme for inclusion of scientific articles

Line 173, 181 The numbers in the Table 1 and in the text are not matching and it should be checked and corrected

Line 374. The used term fluctuations is not appropriated and it should be changed

Kind Regards

Author Response

Point-by-point Response to Reviewer 3

COMMENT

RESPONSE

The author summarized finding of recent meta-analyses about utilization of human serum albumin infusion in critically ill and perioperative patients.

I recommend that this very well written manuscript describing scientifically important theme could be published after only slight minor corrections.

The main strength of the manuscript is strict and appropriate methodological approach that is used during literature search, selection of articles and analysis and interpretation of obtained data.

Thank you very much for your constructive review and positive feedback. We appreciate your recognition of the methodological approach and the scientific importance of the theme. We will certainly address any minor corrections to ensure the manuscript's quality and clarity.

There are several points for suggested minor revision:

1. The modification of title could be considerate to avoid repeating of term review; the readers could become confused if it is narrative review of rapid review; My suggestion for title may be Effectiveness of Clinical Use of Human Albumin in Critically Ill and Perioperative Patients: Rapid Review of Recent Meta-Analyses

We have carefully considered the feedback from both you and the academic editor. To maintain clarity and avoid the impression of a systematic review, we propose the title: "Human Albumin Infusion in Critically Ill and Perioperative Patients: Narrative Rapid Review of Meta-Analyses from the Last Five Years." I believe this title accurately reflects the content and methodology of the manuscript while addressing concerns about potential confusion.

3. Line 36 It may be useful to add that albumin play significant antioxidant roles

The new sentence now reads: “The therapeutic use of albumin is of significant interest because of its unique molecular characteristics, including its substantial antioxidant properties, versatile binding capabilities, and potential for targeted drug delivery.

Line 37 I suggest You to change term molecular science to clinical medicine molecular science

I have excluded the term “molecular science at all and inserted the new sentence given above.

Line 57 Controversy should be changed to controversial

Done.

Line 76. It may be more appropriate to change systematic reviews term with scientific articles as this databases include other types of paper beside systematic reviews too

Done

Section 2.2. I suggest You to rewrite sentences line88-89, line 97-102 in passive voice to avoid using the initials of the authors in the text; the contribution of the author/s is/are usually described only in the paragraph that at the end of the text 

Done

Line 142 The included studies could be changed to selection scheme for inclusion of scientific articles

Done

Line 173, 181 The numbers in the Table 1 and in the text are not matching and it should be checked and corrected

I apologize for this mistake. Text numbers were wrong and are now replaced by the correct data of the table.

Line 374. The used term fluctuations is not appropriated and it should be changed

The term has been changed to “The utilization of HA has experienced variations over time,…